# High-entropy alloys catalyzing polymeric transformation of water pollutants with remarkably improved electron utilization efficiency

Ziwei Yao [1], Yidi Chen [1]✉, Xiaodan Wang[1], Kunsheng Hu[2], Shiying Ren[2], Jinqiang Zhang [2], Zhao Song[3], Nanqi Ren[1] & Xiaoguang Duan [2]✉

High-entropy alloy nanoparticles (HEA-NPs) exhibit favorable properties in catalytic processes, as their multi-metallic sites ensure both high intrinsic activity and atomic efficiency. However, controlled synthesis of uniform multi-metallic ensembles at the atomic level remains challenging. This study successfully loads HEA-NPs onto a nitrogen-doped carbon carrier (HEAs) and pioneers the application in peroxymonosulfate (PMS) activation to drive Fenton-like oxidation. The HEAs-PMS system achieves ultrafast pollutant removal across a wide pH range with strong resistance to real-world water interferences. Furthermore, the nonradical HEAs-PMS system selectively transforms phenolics into high-molecular-weight products via a polymerization pathway. The unique non-mineralization regime remarkably reduces PMS consumption and achieves a high electron utilization efficiency of up to 213.4%. Further DFT calculations and experimental analysis reveal that Fe and Co in HEA-NPs act as the primary catalytic sites to complex with PMS for activation, while Ni, Cu, and Pd serve as charge mediators to facilitate electron transfer. The resulting PMS* complexes on HEAs possess a high redox potential, which drives spatially separated phenol oxidation on nitrogen-doped graphene support to form phenoxyl radicals, subsequently triggering the formation of high-molecule polymeric products via polymerization reactions. This study offers engineered HEAs catalysts for water treatment with low oxidant consumption and emissions.

The treatment of organic wastewater is an increasingly critical issue in the long-term development of society, necessitating the development of environmentally friendly, cost-effective, and efficient water purification technologies[1,2]. Advanced oxidation processes (AOPs) are pivotal in breaking down and mineralizing organic pollutants to

mitigate environmental contamination[3,4]. To meet the rapidly growing demands, the development of high-performance catalysts with high activity, selectivity, and stability of catalytic reactions is essential. Extensive research has produced a variety of catalysts, such as single atoms, bimetallic single atoms, metal oxides/sulfides, and others[5–9].

[1]State Key Laboratory of Urban Water Resource and Environment, Shenzhen Key Laboratory of Organic Pollution Prevention and Control, School of Civil and Environmental Engineering, Harbin Institute of Technology, Shenzhen, Shenzhen, P. R. China. [2]School of Chemical Engineering, The University of Adelaide, Adelaide, SA, Australia. [3]School of Materials and Environmental Engineering, Shenzhen Polytechnic University, Shenzhen, P. R. China.
✉ e-mail: chenyidi@hit.edu.cn; xiaoguang.duan@adelaide.edu.au

There is a consensus that the elemental composition and crystal structure of catalysts significantly influence their catalytic performance. Therefore, the development of more advanced catalysts appears imminent, offering extensive opportunities for microelectronic structure fine-tuning and performance optimization[2,10–12].

High-entropy alloys, a distinctive class of multi-metal alloys typically composed of five or more metals, have garnered widespread research interest in recent years and have found extensive applications in energy conversions and green synthesis[13,14]. The disordered phase of their elements provides inherent advantages, including elemental diversity and structural stability, leading to significantly higher chemical potential and different surface electronic properties compared to traditional single/mixed metal counterparts[15,16]. In catalysis, the adsorption of molecules and intermediate substances on the catalyst surface influences catalytic activity. Compared to pure elements, these adsorption energies can be regulated through alloying to enhance catalytic activity[17]. Consequently, HEAs have been extensively explored in diverse catalytic reactions such as hydrogen evolution, oxygen evolution, alcohol oxidation, and ammonia oxidation, where these alloys demonstrate superior performance compared to traditional catalysts[18–21]. However, the primary challenges associated with high-entropy alloys are the precise synthesis and characterization, as well as complex reaction mechanisms due to the multicomponent structures.

Herein, to enhance the electronic structure of high-entropy alloys for intermediate adsorption, we report the synthesis of CuPdFeCoNi alloy nanoparticles (NPs) loaded onto nitrogen-containing carbon materials using a combination of wet chemistry and controlled pyrolysis methods. The findings reveal that the modified electronic structure accelerates the activation of peroxymonosulfate (PMS) compared to NG for organic pollutant removal from water. The HEAs-PMS system operates through a non-radical electron-transfer pathway (ETP) with high efficiency under harsh pH conditions and resistance to interference from water background factors. Furthermore, a comprehensive analysis of the structure and composition of HEAs was conducted. The associated mechanism was investigated using electron paramagnetic resonance (EPR) and probe experiments, substantiated with electrochemical characterization. To our delight, the HEA-induced non-radical regime yielded a substantial amount of polymerization products on the catalyst surface against mineralization, resulting in an exceptionally high electron utilization efficiency of up to 213.4%. This is attributed to the higher HEAs-PMS* composite potential and faster electron transfer (fast pollutant oxidation kinetics), coordinated by the uniformly distributed multi-elements in HEAs, producing high-polymerization-degree products. The long-term stability and practicality of HEAs were demonstrated through a 50-day continuous flow and actual wastewater removal efficiency experiment. Overall, this study presents a solution for developing advanced heterogeneous catalysts suitable for water remediation with reduced oxidant consumption and upcycled water pollutants.

## Results

### Synthesis and characterization of HEAs

A straightforward one-pot oil-phase synthesis approach was adopted by heating five metal salt precursors (Cu, Pd, Fe, Co, and Ni) at 220 °C for 2 h, followed by thermal decomposition along with a carbon precursor to obtain HEAs[12,22]. The calculated composition of the obtained HEAs was determined to be $Cu_{12}Pd_{11}Fe_{10}Co_{11}Ni_{12}$ through inductively coupled plasma-optical emission spectrometry (ICP-OES) and elemental analysis (Supplementary Tables 1 and 2). The synthesis procedure, depicted in Fig. 1a and detailed in the Experimental section, successfully yielded uniform $Cu_{12}Pd_{11}Fe_{10}Co_{11}Ni_{12}$ nanoparticles with a diameter of $8.5 \pm 0.8$ nm, as shown in transmission electron microscopy (TEM) images (Fig. 1b and Supplementary Figs. 1 and 2). The powder X-ray diffraction pattern confirmed the face-centered cubic (*fcc*) crystalline structure of CuPd alloy (JCPDS No. 48–1551) NPs, with

the main phase identified as CuPd alloy by the peaks at around 41.4° and 48.2°, corresponding to the (111) and (200) facets (Supplementary Fig. 3)[23]. Additionally, the pronounced shift in the position of the broad diffraction peak, in contrast to the diffraction peaks of pure metals (Pd, Ni, Fe, Co, and Cu), indicates a significant alteration. The subtle peak shift for the CuPd alloy further suggests the incorporation of Fe, Ni, and Co into the CuPd lattice, leading to the creation of a uniform alloy structure with concurrent lattice contraction[24]. High-angle annular dark-field scanning TEM (AC-HAADF-STEM) image in Fig. 1c shows a lattice spacing of 0.217 nm, corresponding to the (111) facet of $Cu_{12}Pd_{11}Fe_{10}Co_{11}Ni_{12}$ NPs (Supplementary Fig. 4). Moreover, AC-HAADF-STEM-EDS elemental mappings show that each element is uniformly distributed, demonstrating that the five metals (Pd, Fe, Co, Ni, and Cu) are evenly positioned on the surface of HEA NPs (Fig. 1d)[25]. These comprehensive characterizations substantiate the successful synthesis and loading of $Cu_{12}Pd_{11}Fe_{10}Co_{11}Ni_{12}$ NPs onto N-doped graphene (NG).

To determine the electronic and coordination structures of Cu atoms in HEAs, X-ray absorption near-edge spectroscopy (XANES) and extended X-ray absorption fine structure (EXAFS) were utilized. XANES curve reveals that the white line intensity is higher than Cu foil but much lower than that of CuO and CuPc, confirming that the Cu species in HEAs are mainly present in the metallic state (Fig. 1e)[24,26]. Moreover, the pre-edge centroid of HEAs is situated between the Cu foil and CuPc, indicating potential coordination between Cu atoms and N in NG. The Fourier-transformed extended X-ray absorption fine structure (FT-EXAFS) spectrum of HEAs in R space displays two dominant peaks located at approximately ≈1.7 Å for the Cu-N bond and a peak offset at 2.6 Å, corresponding to the Cu-Cu interaction. This offset indicates the presence of Cu-M interactions (M = Pd, Fe, Ni, Co) within the HEAs, which is likely caused by the mutual interaction of multiple elements for HEAs (Fig. 1f). Besides, the wavelet transforms (WT) of the $k^3$-weighted EXAFS spectra of HEAs also reveal two dominating peaks maximum at ≈4–10 Å$^{-1}$ ascribed to Cu−N bonding and Cu-M interactions (Fig. 1g). Strong Cu-Cu and Cu-N coordination can be observed in the WT contour plots of Cu foil and CuPc, which agrees with the results from EXAFS spectra. Additionally, X-ray absorption spectroscopy of Ni atoms in HEAs also yields similar conclusions, despite the dominant Ni-M interactions (Supplementary Fig. 5). This further corroborates the earlier characterization findings, confirming the incorporation of Ni into the CuPd alloy.

### Superior catalytic efficiency of HEAs

The catalytic capability of HEAs was investigated by using PMS as a peroxide precursor, and the Fenton-like activity was assessed by employing phenol as a model organic contaminant. As shown in Fig. 2a, minimal adsorption occurred in the catalyst-only systems, with the HEAs system achieving 100% phenol removal in 10 min with PMS addition, while the NG system exhibited less than 60% phenol removal. Notably, the degradation rate of HEAs exceeded that of NG by over 7 times (Supplementary Fig. 6), suggesting that the integration of alloy NPs sites into NG could be crucial active sites for PMS activation[6]. Additionally, the optimization of PMS concentration demonstrated that complete degradation was achievable with a PMS dose of 0.25 mM, following a positive correlation of oxidation kinetics with PMS concentration, likely due to its positive feedback effects on PMS activation (Supplementary Fig. 7). The removal rates decreased with increasing initial phenol concentrations, though complete phenol removal was still achieved within 10 min at a concentration of 0.1 mM (Supplementary Fig. 8). However, when the initial phenol concentration exceeded the electron equivalents that PMS could accept (0.25 mM), the removal efficiency was significantly reduced.

Cobalt, iron, and other metal ions have been discovered to exhibit catalytic performance as classic activators in Fenton-like reactions, widely utilized for PMS activation. However, the efficiency of

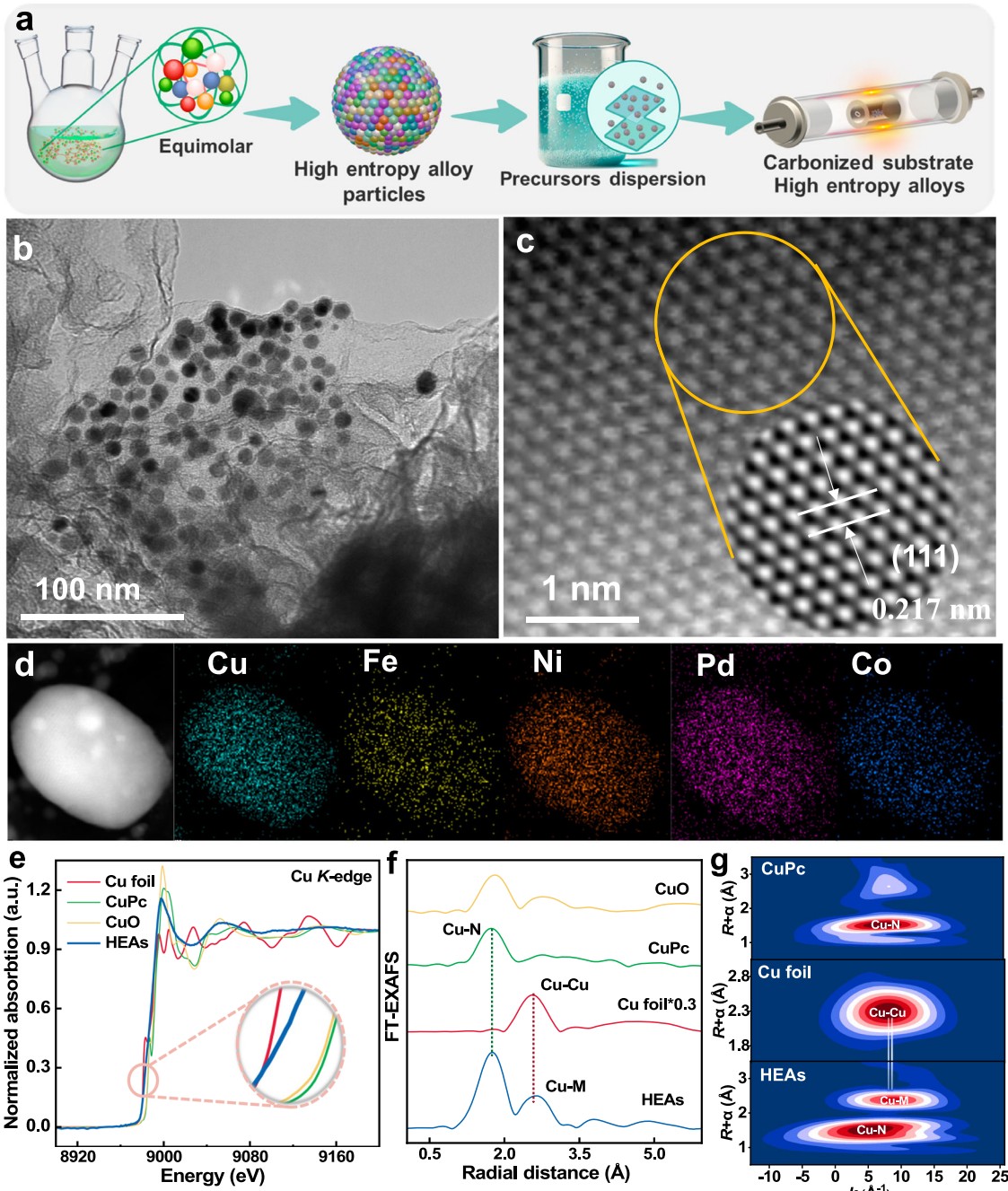

**Fig. 1 | Synthesis and characterization of HEAs. a** The schematic diagram for synthesis process of HEAs. **b** TEM images of HEAs. **c** Aberration-corrected HAADF-STEM image and the corresponding inverse FFT pattern of HEAs (highlighted in the yellow circle). **d** Elemental mapping for HEAs (scale bar, 5 nm). **e** XANES at Cu L-edge. **f** FT-EXAFS spectra in R space. **g** WTs of EXAFS spectra of HEAs, CuO, Cu foil, and CuPc.

heterogeneous catalysts in comparison to metal ions such as $Co^{2+}$ remains limited[7]. To investigate the exceptional performance of HEAs in phenol degradation, experiments were conducted using equimolar concentrations of metal ions ($Co^{2+}$, $Fe^{3+}$, $Ni^{2+}$, $Pd^{2+}$, and $Cu^{2+}$) as comparative catalysts based on the metal content in HEAs (Supplementary Fig. 9 and Table 3). It was observed that at a PMS concentration of 0.25 mM, only $Co^{2+}$ exhibited a significant degradation performance. Further degradation experiments were conducted at higher concentrations using cobalt ions and a mixture of 5 other metal ions. The results in Supplementary Fig. 10 showed that even with a 40-fold increase in $Co^{2+}$ and a 20-fold increase in mixed ions, the phenol removal efficiency still cannot compete with the HEAs-PMS system. The apparent kinetic constant of the HEAs-PMS system was 5.3 times

and 2.7 times higher than that of the homogeneous $Co^{2+}$–PMS and mixed ions-PMS systems, respectively. These results further support the advantages of high-entropy alloy NPs-loaded catalysts in terms of high intrinsic activity and low metal leaching in addressing environmental pollution. Furthermore, monitoring the PMS consumption rate via the iodine reduction reaction suggested that the exceptional performance of HEAs could be attributed to remarkably enhanced PMS activation on the alloy NPs surface, achieving a rate 12 times higher than NG (Supplementary Fig. 11)[27].

The impacts of the co-existing dissolved organic matter (DOM, represented by humic acid) and typical anions species (e.g., $Cl^-$, $SO_4^{2-}$, $HCO_3^-$, $NO_3^-$, and $H_2PO_4^-$) on phenol removal by HEAs and NG were further investigated. As illustrated in Fig. 2b, DOM at the concentration

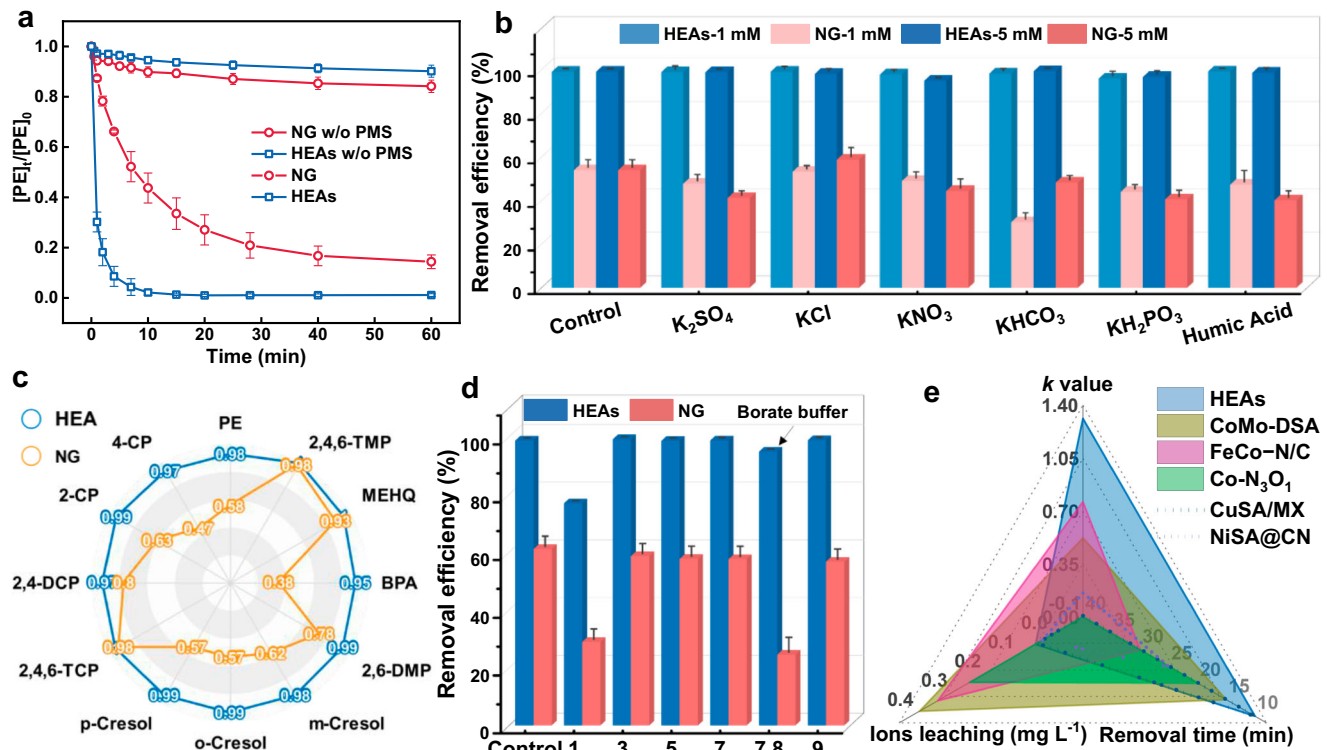

**Fig. 2 | Catalytic Efficiency of HEAs. a** The removal efficiency against organic contaminants of NG and HEAs catalysts. **b** HEAs and NG catalytic removal efficiency of phenol with interference of 5/10 mM salts and humic acid. **c** Evaluate the removal efficiency of different pollutants in the HEAs-PMS system and the NG-PMS system within 15 min. **d** HEAs and NG catalytic removal efficiency of phenol under different pH and borate buffer environments. **e** Recently reported performance comparisons of different catalysts with HEAs. Error bars represent the standard deviation, obtained by repeating the experiment three times. Dosage: PMS: 0.25 mM, reaction solution: 50 mL, [Pollutants]$_0$: 0.1 mM, catalyst: 0.1 g L$^{-1}$, reaction time: 60 min.

of 5 and 10 mg L$^{-1}$ has nearly no impact on phenol removal, with the removal efficiency over 99.7% by HEAs. Additionally, the existence of Cl$^-$, SO$_4^{2-}$, HCO$_3^-$, NO$_3^-$, and H$_2$PO$_4^-$ at concentrations of 5 and 10 mM showed negligible influence on phenol removal. In contrast, NG exhibited notable effects on the removal process (detailed in Supplementary Fig. 12). In traditional Fenton reactions, primary reactive oxygen species (ROS), such as ·OH and SO$_4^{·-}$, would readily interact with various anions and affect the removal process; thus, that the HEAs-PMS and NG-PMS systems may follow a non-radical pathway[28]. Our investigation into the oxidation capacity of HEAs revealed faster oxidation kinetics of the HEAs-PMS system, when purifying 11 other substituted phenols (Fig. 2c and Supplementary Fig. 13). Similar phenomenon was observed upon oxidizing other organic substitution models with electron-donating groups (methyl and methoxy), with the kinetics accelerating as the electron-donating ability increased, implying that the oxidation was dominated by the nonradical ETP. (Supplementary Fig. 14)[7].

To assess the stability and anti-interference efficacy of HEAs in practical water treatment scenarios, the degradative performance of the HEAs-PMS system was evaluated under harsh pH conditions (Fig. 2d). The pH of the reaction solution had a minor impact on the catalytic activity of HEAs, with the reaction rate remaining unaffected when using a borate buffer (Supplementary Fig. 15). Despite a slight decrease at pH 1, HEAs showed a distinct contrast to NG, whose catalytic activity significantly diminished under borate and strong acid conditions. This disparity could be attributed to the acidic environment affecting PMS, hindering activation and decomposition efficiency, consequently affecting removal performance. In conclusion, the Fenton-like reaction, primarily facilitated by the HEAs-PMS system, greatly overcomes the limitation of a narrow pH application range. To address the risk of secondary

pollution from metal leakage in metal-based catalysts, ICP was employed to monitor the leakage of metal ions in heterogeneous Fenton-like reactions (Supplementary Table 4). A comparison of the ion leakage levels of Co$^{2+}$, Fe$^{3+}$, Ni$^{2+}$, Pd$^{2+}$, Cu$^{2+}$ from HEAs catalyst with reported levels from single-atom metal-based catalysts in the literature revealed that the levels were significantly lower and nearly negligible. In particular, the HEAs catalyst outperformed previous reported heterogeneous catalysts, showing a greater normalized $k$ value and removal performance (Fig. 2e and Supplementary Table 5)[4,7,29–31]. These results collectively demonstrate the high performance and potential of HEAs in environmental catalysis.

## Mechanism study on the Fenton-like activity of HEAs

To elucidate the impact of ROS on the HEAs-PMS system, quenching experiments were conducted to validate ROS generation and discern their contribution to phenol oxidation. Methanol (MeOH) and tert-butyl alcohol were employed as quenchers for ·OH/SO$_4^{·-}$ ($9.7 \times 10^8$ M s$^{-1}$, $2 \times 10^7$ M s$^{-1}$) and ·OH ($6.0 \times 10^8$ M s$^{-1}$), respectively[32]. As depicted in Fig. 3a, neither of these scavengers exhibited significant inhibitory effects in the HEAs-PMS and NG-PMS systems. Benzoic acid and nitro-tetrazolium blue chloride (NBT) were utilized as probes to further confirm the involvement of ·OH and O$_2^{·-}$ in the oxidation process[33]. As a result, benzoic acid removal was almost negligible, and neither monoformazan nor diformazan was identified, further substantiating the minimal impact of ·OH and O$_2^{·-}$ (Supplementary Figs. 16 and 17). Figure 3b revealed the absence of electron spin resonance capture signals for ·OH, SO$_4^{·-}$, and $^1$O$_2$ in both HEAs-PMS and NG-PMS systems. Meanwhile, anaerobic experiments revealed the trivial impact of dissolved oxygen, suggesting that oxygen does not participate in the oxidation reactions in both the HEAs-PMS and NG-PMS systems (Supplementary Fig. 18). Moreover, the addition of 2,2,6,6-

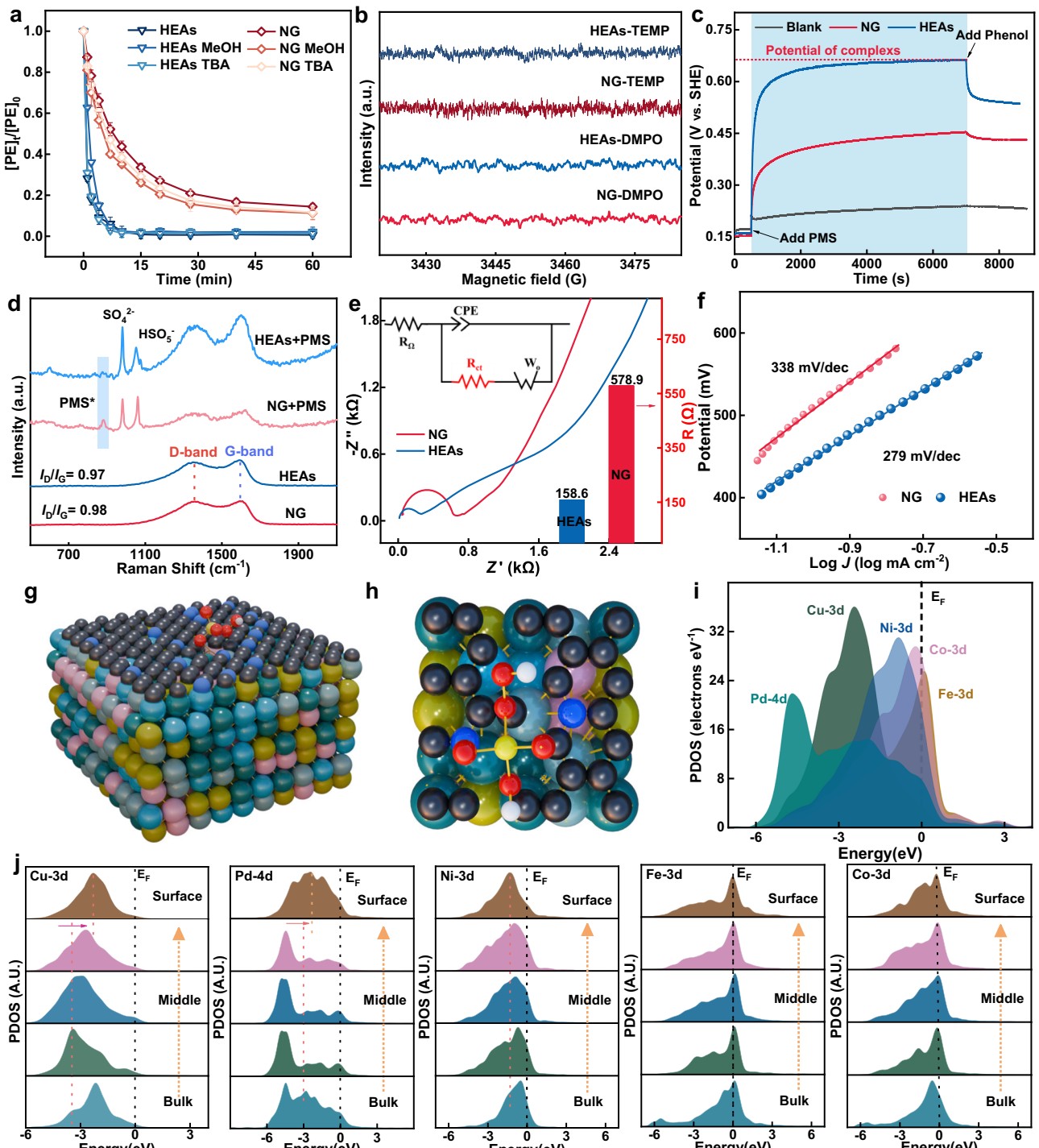

**Fig. 3 | Identification of active species. a** Quenching effects by various scavengers. The error bars represent the standard deviations from triplicate tests. Dosage: [Phenol]$_0$: 0.1 mM, PMS: 0.25 mM, scavengers: 2.5 mM, reaction solution: 50 mL, catalyst: 0.1 g L$^{-1}$. **b** EPR tests were performed on DMPO and TEMP adducts within the HEAs-PMS system during PMS activation. **c** The open circuit potential on HEAs–GCE and NG–GCE electrode. **d** In situ Raman spectra of HEAs-PMS and NG-PMS in the liquid solution. **e** Nyquist plots and fitting calculations of internal resistance for HEAs and NG. **f** Tafel slope on the HEAs−GCE and NG−GCE electrodes in phenol solutions. **g** The side view of the structural configuration of PMS on HEAs. **h** The top view of the structural configuration of PMS on HEAs. **i** The PDOSs of the HEAs. **j** The site-dependent PDOSs of Cu, Pd, Ni, Fe, and Co in HEAs.

tetramethylpiperidine (TEMP) as a $^1O_2$ quencher did not affect phenol removal efficiency, signifying the negligible influence of $^1O_2$ on the catalytic performance (Supplementary Fig. 19). Simultaneously, substituting $D_2O$ for deionized water in the solution did not facilitate the oxidation kinetics, and conducting experiments with 9,10-Diphenyl-lanthracene (DPA) as a probe did not generate the featured product from $^1O_2$ oxidation (Supplementary Figs. 20 and 21)[34]. These results

indicated that $^1O_2$ did not present in the HEAs-PMS and NG-PMS systems. The negligible inhibitive impact of dimethyl sulfoxide as a specific scavenger suggests the absence of high-valence metal-oxo species in the HEAs-PMS system (Supplementary Fig. 22)[35]. These collective findings exclude the contributions of radicals ($^•OH$ or $SO_4^{•-}$), $^1O_2$, and high-valence metal-oxo species, implying that PMS-triggered ETP are likely involved in phenol removal in a non-radical manner.

Various electrochemical techniques were employed to assess ETP-mediated phenol oxidation in the HEAs-PMS and NG-PMS systems, including OCPT, LSV, CA, EIS, Tafel curve, and ECSA analysis[28]. As shown in Fig. 3c, after the injection of PMS, OCPT promptly increased for HEAs-GCE and NG-GCE, indicating the formation of surface-activated PMS complex (PMS*) which elevated the catalyst potential. Notably, the potential of HEAs-PMS* exceeded that of NG-PMS*, implying a higher oxidation capacity. Upon phenol addition, a subsequent decrease in OCPT indicated ETP occurs between HEAs/NG-PMS* and phenol. Moreover, in situ Raman spectroscopy in Fig. 3d directly evidenced the existence of the PMS* reactive complex on HEAs and NG surfaces.

Further monitoring of PMS content during the OCPT experiment revealed the more efficient interaction of alloy NPs with PMS, giving rise to the higher potential observed for the HEAs−GCE electrode (Supplementary Fig. 23). The CV experiment showed significantly higher current density for phenol oxidation by HEAs than NG, confirming the superior electron transfer capability of alloy NPs from the pollutant to PMS (Supplementary Fig. 24). The proposition was also supported by the CA, current−time and ECSA measurement (Supplementary Figs. 25–27). The conductivity and electrochemical kinetics of the HEAs were further estimated using EIS and the *Tafel* slope, defining the rate-limiting step (Fig. 3e and Supplementary Fig. 28). The circuit fitting results indicated that the impedance of HEAs (158.6 Ω) was significantly lower than that of NG (578.9 Ω), due to the presence of metallic HEAs, which benefit the charge migration processes. *Tafel* slopes were obtained by fitting LSV data to the *Tafel* curve, resulting in calculated Tafel slopes of 279 and 338 mV dec$^{-1}$ for HEAs and NG, respectively (Fig. 3f). These disparities suggest that HEAs exhibit faster oxidation kinetics and higher electron transfer capacity at high potentials compared to NG. Galvanic reactor experiments were performed to analyze the mechanism of PMS activation by HEAs/NG as shown in Supplementary Fig. 29. The electrode potential within the PMS chamber was notably higher than that in the phenol chamber, indicating that the PMS chamber served as the cathode chamber, while the phenol solution acted as the anode chamber (Supplementary Fig. 30). Consequently, phenol was directly oxidized through proton-coupled ETP in the galvanic cell reaction and synchronously mediated electron flow from the organic cell to the surface-activated PMS on the other end. The recorded current changes during the galvanic reactions demonstrated that HEAs could generate a significantly larger current compared to NG, underscoring the superior PMS activation (electron acceptance) and phenol oxidation (charge donation) kinetics on HEAs-NPs via an ETP regime[34].

To unravel the superior activity and electron transfer capability of HEAs, the projected density of state (PDOS) for PMS adsorption on HEAs was calculated. Initially, the PDOS of three HEAs with slight stoichiometric differences was compared. As shown in Supplementary Fig. 31, the comparison models displayed highly similar electronic structures with minimal changes in the peak positions and patterns of d orbitals for each element. Therefore, slight variations in HEAs stoichiometry are unlikely to significantly affect the electronic structure in the PDOS results[22,25]. Subsequently, the HEAs structure with the highest stability was selected as the lattice model (Fig. 3g and h). To further understand the electronic structure, Fig. 3i has illustrated the PDOS of each element in the HEAs. Notably, the Pd-4d orbitals occupied the deepest position near the valence band ($E_V = 0$ eV) at −4.6 eV, serving as an electron reservoir. The Ni-3d orbitals exhibited a peak at Ev −1.0 eV, while the Co-3d orbitals showed a pronounced intersection with high electron density at the Fermi level ($E_F$), supporting the high electroactivity of Ni and Co sites to coordinate surface redox reaction. Additionally, the significant overlap between different d orbitals indicated strong bonding and electronic communication between the metal elements. These strong interactions enabled the coupling of reactants (PMS) with the 3d orbitals of Fe and Co sites, facilitating

flexible electron modulation and accelerating electron transfer on the surface of the HEAs NPs. This signified that the 3d orbitals of Cu, Ni, Co, and Fe also acted as bridging intermediaries, helping to alleviate the energy barrier for ETP during oxidation processes[36].

To further identify the electronic modulations, the site-dependent PDOSs of each element were supplied in HEAs (Fig. 3j). From bulk to surface sites, an evident downward shift was observed for Cu-3d and Pd-4d orbitals, while Pd sites displayed an alleviation of the $e_g–t_{2g}$ splitting effect, supporting an enhanced electron transfer efficiency. For Ni sites closer to the surface, the 3d orbitals showed a slight shift to support the electron transfer activity of the HEAs. Both Fe and Co sites exhibited a relatively stable d-band center, attributed to a dual anchoring effect from the shielding effect of neighboring metal orbitals. This helped maintain the electron boosting center and facilitated coupling with PMS, exhibiting the strong electron-donating capability for the oxidation process. Due to low barriers for electron depletion, Co, Fe, and Ni sites were identified as buffer sites, protecting the metallic and electroactive Cu and Pd sites within HEAs NPs from oxidation. This ensures superior removal performance and stability of HEAs in ETP oxidation. Based the aforementioned results, we inferred that Fe and Co act as the primary reactive sites for binding with PMS, and meanwhile, Ni, Cu, and Pd are mediators that accelerate electron transfer to phenol adsorbed on the surface of nitrogen-doped graphite, thereby enhancing the kinetics of phenol oxidation.

## Mechanistic study of polymerization in pollutant removal

Recently, catalytic systems based on metal oxides and carbonaceous materials have been reported to induce polymerization of specific aromatic pollutants (phenolics and anilines) with the presence of persulfates[37–39]. Through the polymerization pathway, organic pollutants are converted from aqueous solutions into recyclable products deposited onto the catalyst surface, thus altering the AOPs pathway from mineralization to polymerization. Such regime transition would remarkably reduce the peroxide inputs with high decontamination efficiency and minimize the carbon emission in AOPs. Here, thermogravimetric analysis (TGA) was conducted to examine the presence of phenol polymerization products on the surfaces of these systems. The TGA results revealed more significant pyrolysis decomposition of polymeric products for HEAs after the AOPs treatment (15.6%) compared to the pristine HEAs and NG, manifesting as mass losses of 9.9% and 4.5%, respectively (Fig. 4a and Supplementary Fig. 32). This observation suggested that HEAs exhibited a superior potential to promote polymeric transformation of pollutants, stemming from the more favorable PMS* complex potential and faster electron transfer capability to rapidly convert organic pollutants into monomer radicals via electron abstraction to initiate the polymer chain reaction.

Additionally, the total organic carbon (TOC) removal efficiency of the HEAs-PMS and NG-PMS systems was analyzed to shed light on the discrepancy in electron equivalents observed during phenol mineralization (Supplementary Table 6). The results revealed that while the theoretical maximum electron equivalents obtained by PMS (0.5 mM) were significantly lower than the actual electron equivalents lost during phenol mineralization (1.067 mM)[40,41]. This disparity could be attributed to the selective oxidation of phenol via one-electron transfer in the polymerization pathway (PMS:phenol = 1:2), rather than the mineralization process, which requires much higher theoretical PMS consumption (PMS:phenol = 14:1). Benefiting from this oxidative pathway, the HEAs-PMS system (213.4%) demonstrated higher electron utilization efficiency compared to the NG-PMS system (146.7%), significantly reducing the consumption of oxidants in practical applications compared to homogeneous mineralization catalysis (Fig. 4b).

Subsequently, we use BHT and Ferulic acid (FA) as a polymerization inhibitor to suppress polymerization activity via blocking the phenoxyl radical intermediate. The addition of BHT(2,4,6-tri-tertbutylphenol) resulted in a notable reduction in phenol removal,

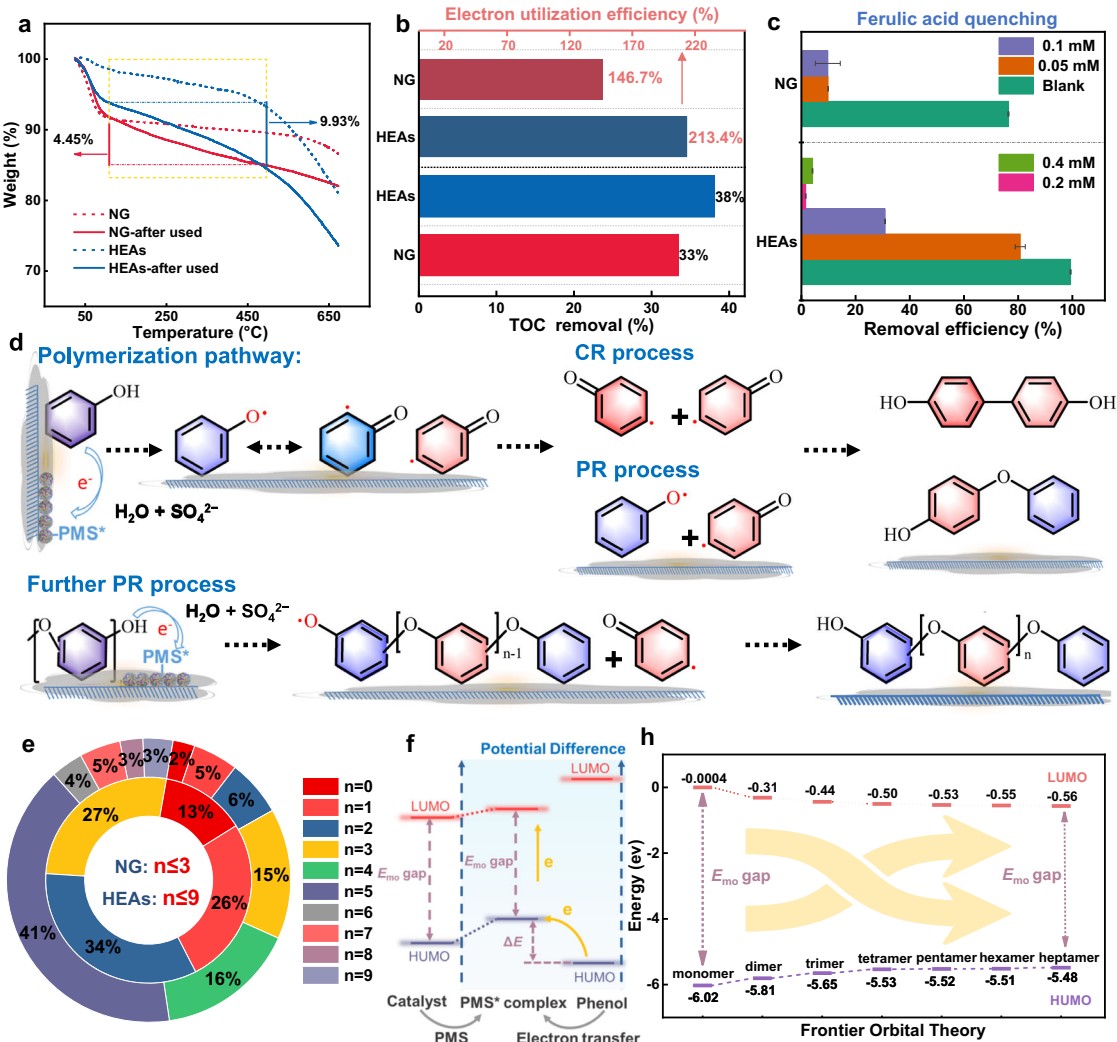

**Fig. 4 | Characteristics of the polymerization products. a** TGA curves of HEAs and NG before and after the reaction. **b** HEAs-PMS and NG-PMS aqueous solution systems final TOC removal efficiency, and electron utilization. **c** The effect of different concentrations of FA on the phenol removal efficiency in the NG-PMS and HEAs-PMS systems (The error bars represent the standard deviations from triplicate tests). **d** Proposed reaction pathways of the oxidative coupling and polymerization of phenol on the catalyst surface. **e** The distribution of different units polymerization-degree products on the surface of NG and HEAs catalysts. **f** Schematic diagram of the frontier orbital theory for catalytic polymerization of phenol on the catalyst surface. **h** Calculated frontier orbital diagrams for different units polymerization-degree phenol. Dosage: [Phenol]$_0$: 0.1 mM, PMS: 0.25 mM, reaction solution: 50 mL, catalyst: 0.1 g L$^{-1}$.

demonstrating the presence of polymerization processes in both HEAs-PMS and NG-PMS systems (Supplementary Fig. 33). FA can convert phenoxyl radicals back to redox-inert phenol precursor, making it a more precise phenoxyl radical scavenger (Fig. 4c and Supplementary Fig. 34). The results show that ca. 0.05 mM FA completely inhibit phenol removal in the NG-PMS system, while in the HEAs-PMS system, the inhibition degree increased with FA concentration, eventually reaching complete inhibition (above 0.2 mM). Thus, these results firmly support that the higher potential of the PMS* complex in the HEAs-PMS system (compared to NG-PMS) would induce faster generation and accumulation of phenoxyl radicals, thereby accelerating the polymerization process.

Analysis of the intermediates/products was conducted to investigate the phenol transformation pathways in both systems. The ultrahigh performance liquid chromatograph-mass spectrometer (UPLC-QTOF-MS) was employed to examine the intermediate products of phenol in the aqueous phase and adsorbed on the catalyst surfaces[34]. We used the nucleophile acetonitrile as a scavenger, which ruled out the double-electron-transfer process generating the phenoxonium ion as the intermediate (Supplementary Fig. 35). Thus,

phenol in the solution first donates one electron to the PMS* complex to form phenoxyl radicals with various resonance states (Fig. 4d). These phenoxyl radicals serve as precursors for oligomeric products in the radical chain reactions, further undergoing C−C/C−O coupling to generate dimeric products with different coordination (same $m/z = 185$).

Notably, dimers and other products may also lose electrons to form larger organic radicals, increasing the polymerization degree via coupling dimer radicals or other polymer radicals to form high-molecule polymers. The generated polymers repeatedly participate in oxidative coupling reactions, providing opportunities for pairing any two polymer radicals together, thereby increasing the polymerization degree and leading to a large quantity of high-molecular-weight polymers adhering to the catalyst surface[39]. In Supplementary Figs. 36 and 37, further analysis of the polymerization products on the catalyst surface in the HEAs-PMS and NG-PMS systems revealed the presence of polyphenol products with a higher polymerization degree on the surface of HEAs ($m/z = 1013$, $n = 9$), while only lower-molecular polymerization products were detected on the NG surface ($m/z = 461$, $n = 3$). Further analysis of extraction and concentrated bulk solution

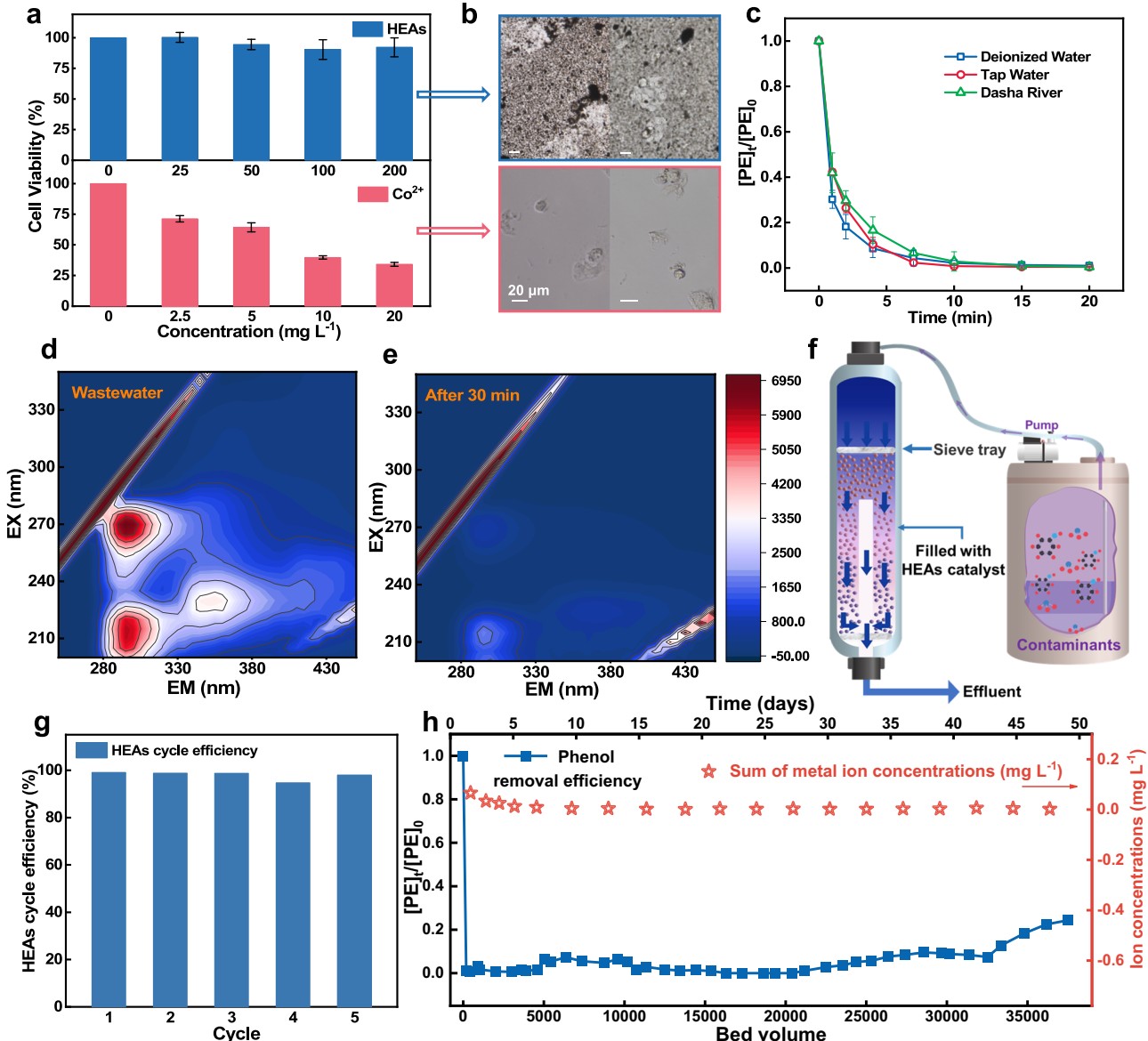

**Fig. 5 | Assessment of the adaptability and stability of HEAs. a** Cell viability of 16HBE under different concentrations of HEAs and $Co^{2+}$. Cell culture time: 48 h. **b** Cells in the culture medium were observed under a microscope. **c** Treatment of phenol in real water matrices. **d, e** EEM spectra of collected coal chemical industry wastewater before (**d**) and after treatment (**e**) by HEAs-PMS system. **f** Schematic of the fixed-bed column continuous flow system. **g** The phenol removal efficiency of HEAs in cyclic experiments. **h** The efficiency of phenol removal and the metal leakage of HEAs in a continuous flow system. Error bars represent the standard deviation, obtained by repeating the experiment three times. Dosage: $[Phenol]_0$: 0.1 mM, PMS: 0.25 mM, reaction solution: 50 mL, catalyst: 0.1 g $L^{-1}$.

from the HEAs-PMS and NG-PMS systems revealed the detection of products ranging from dimers to tetramers ($m/z = 369$, $n = 2$), as well as partially oxidized ring-opening products, indicating that the lower-polymerization-degree products were dissolved into the solution (Supplementary Figs. 38–44).

Moreover, upon the depletion of the electron source (PMS), the insufficient coupling of polymer radicals can lead to the termination of polymerization reactions. This result is consistent with the findings of the TGA experiment that HEAs exhibit higher pollutant-to-polymer transformation efficiency. In Fig. 4e, polymerization degree distribution analysis on the surfaces of NG and HEAs revealed that tetramers ($n = 2$) and pentamers ($n = 3$) are predominantly observed on the NG surface, while most hexamers ($n = 4$) to nonamers ($n = 7$) are present on the HEAs surface. In contrast, lower-polymerization-degree products are dissolved in the solution phase. Thus, the higher redox potential of HEAs-PMS* and faster electron transfer capability can trigger the rapid generation and accumulation of phenoxyl radicals, thereby enhancing the phenol-to-polymer transformation kinetics to yield larger polyphenol products.

Further verification of the above mechanism was conducted through the frontier orbital theory, as shown in Fig. 5f, h and Supplementary Fig. 45), the energy gaps between the highest occupied molecular orbital (HOMO) and the lowest unoccupied molecular orbital of polyphenol with different molecular sizes decrease gradually with the increased polymerization degree. This suggests that reactants with higher degrees of polymerization have a higher HOMO and lower energy gap (ΔE) for losing electrons, making them more easily oxidized into phenoxy radicals and form the corresponding polymeric products. Additionally, the energy barriers for the formation of organic radicals in products with different degrees of polymerization indicate that as the degree of polymerization increases, the reaction becomes more thermodynamically favorable (Supplementary Fig. 46).

To better understand why HEAs achieved higher degrees of polymerization, phenol polymeric products in different equivalents were prepolymerized on the surfaces of HEAs and NG using a PMS-triggered polymerization reaction. As shown in Supplementary Fig. 47, after introducing different amounts of phenol polymerization products onto HEAs, these surfaces continued to rapidly activate PMS, resulting in further polymerization of phenol and the production of high-molecular-weight products. In contrast, NG quickly lost its ability to activate PMS. This phenomenon further demonstrates that the conjugated aromatic structure and van der Waals interactions of nitrogen-doped graphite preferentially adsorb and enrich polymerization products, which predominantly form on the graphite surface. Meanwhile, the separation of active sites enables PMS activation on the HEA NPs, sustaining the ETP-mediated polymerization process. In contrast, the polymerization products in NG indiscriminately covered the reactive sites for PMS activation, leading to a rapid loss of catalytic activity. Simultaneously, OCPT experiments in Fig. 3c showed that HEAs maintained a higher complex potential than NG with the presence of phenol, allowing for the continued polymerization process. These conclusions clearly indicate that the separation of active sites for PMS and organic pollutants in HEAs enables continuous PMS activation, generating a higher potential and maintaining the ETP-mediated polymerization process. At the same time, the graphite interface consistently adsorbs, concentrates, and stabilizes the polymers, leading to the formation of larger polyphenol products with a higher degree of polymerization compared to NG.

### Biotoxicity universality and application stability test

In assessing the suitability of HEAs as catalysts for water treatment, understanding their biotoxicity is crucial. To assess the exposure risk during practical catalyst use, we conducted in the vitro cytotoxicity experiments on human bronchial epithelial cells (16HBE)[7]. As represented by Fig. 5a, CTL assay experiments reveal dose-dependent biological toxicity of $Co^{2+}$ and mixed ions, with cell viability dropping below 35% and 60% at concentrations of $20\,mg\,L^{-1}$ for $Co^{2+}$ and $10\,mg\,L^{-1}$ for the mixed ion concentration (approximately equivalent to HEAs performance) (Supplementary Fig. 48). Notably, the addition of HEAs yielded promising results, maintaining 16HBE cell line viability above 90% even with twice the concentration. This contrast between HEAs and ions was further emphasized in Fig. 5b, where a noticeable difference in cell count and survival morphology in the microscopic field. In addition, the toxicity of the polymeric products was assessed using the Toxicity Estimation Software Tool, which showed a slight increase in toxicity as the polymerization degree increased. This finding highlights the importance of developing efficient recovery technologies to reduce potential environmental impacts and promote the reutilization the upcycled polymeric products. (Supplementary Fig. 49)

Furthermore, in practical water matrices experiments, the HEAs-PMS system demonstrates excellent anti-interference capability in environments with more complex water matrices (Fig. 5c). The secondary wastewater collected from the coal chemical industry was utilized to assess the practicality of HEAs-PMS system. The excitation emission matrix (EEM) spectra shown in Fig. 5d, e indicated that soluble microbial byproducts and phenolic humic acid-like organic compounds in the wastewater samples were easily removed, as evidenced by the significant reduction in fluorescence intensity at the detected peaks.

The catalyst also exhibited excellent stability in the cyclic removal of phenol, achieving a 95% removal of phenol in five cycles (Fig. 5g and Supplementary Fig. 50). In contrast, continuous-flow long-term running experiments provided a more comprehensive evaluation of the catalyst's performance in a fixed-bed reactor for degrading organic pollutants (Fig. 5f)[42]. The results of the 50-day long-term degradation experiment under conditions of $10\,mg\,L^{-1}$ phenol demonstrated that the degradation performance of the HEAs-PMS system gradually declined only after treating a phenol solution with a bed volume of 30,000 (Fig. 5h and Supplementary Fig. 51). Additionally, the detected leakage of metal ions is almost zero (Supplementary Table 7), further proving the robust structural stability of HEAs and sustained removal performance in long-term operation[43].

## Discussion

In summary, we successfully synthesized and applied HEAs for aqueous pollutant removals in the PMS-AOPs system. Compared to NG, the supported HEAs are more effective in catalyzing PMS activation to achieve rapid phenol removal (PMS:phenol = 2.5:1), exhibiting exceptional stability even under harsh pH conditions and immunity to interference from ionic substances in the water background. HEAs exhibited stable and long-lasting catalytic activity in the 50-day long-term experiments with no observable metal leaching. The HEAs NPs secured both high PMS activation activity and excellent conductivity, resulting in enhanced PMS* composite potentials and accelerated electron transfer rates. As a result, the nonradical ETP regime trigger non-mineralization phenol removal via one-electron oxidation and achieved a high electron utilization efficiency of up to 213.4%, leading to the sustained production of high-molecular-weight polyphenolic products. Additionally, due to the environmental friendlies and low biotoxicity of HEAs, this study provides a perspective for designing atom-precision heterogeneous catalysts for the treatment of industrial wastewater with low oxidant consumption and carbon emission.

## Methods
### Preparation of catalysts

In a typical synthesis of HEAs, $Pd(acac)_2$ (7.6 mg), $Ni(acac)_2$ (6.4 mg), $Fe(acac)_3$ (8.8 mg), $Co(acac)_3$ (8.9 mg), $Cu(acac)_2$ (6.5 mg), glucose (60 mg), and CTAC (50 mg) were dissolved in 10 mL of oleylamine, followed by ultrasound treatment for 30 min. Then, the mixture was heated in an oil bath at 220 °C for 2 h. The cooled product was collected by centrifugation and washed three times with a 1:5 volume ratio ethanol/cyclohexane mixture. Finally, the black colloidal products were dispersed in cyclohexane, $C_3N_4$ (500 mg) was added, followed by sonication for 60 min, centrifugation, and washing three times with an ethanol/cyclohexane mixture. After vacuum drying at 60 °C for 12 h, 500 mg of glucose was added and finely ground until a homogeneous dispersion was achieved. The mixture was then transferred into a tubular furnace and annealed at 800 °C for 2 h at a heating rate of 5 °C $min^{-1}$ under a protective Ar atmosphere. Subsequently, it was promptly removed and allowed to cool to room temperature to obtain HEAs. For the synthesis of NC, all the conditions are similar to those of HEAs except for the absence of black colloidal products during the annealing process.

### Catalyst characterization

The high-angle annular dark field scanning transmission electron microscopy (HAADF-STEM) technique using a Titan Themis 80–200 electron microscope (FEI Company, Netherlands) was used to observe the morphology and structure of the catalysts and elemental mapping. The crystal phases of the catalysts were characterized by X-ray diffraction patterns with CuK-α radiation (SmartLab 3kw, Rigaku Co.). Electron capture by 2,2,6,6-Tetramethyl-4-piperidone hydrochloride (TEMP) and radical trapping by 5,5-dimethyl-1-pyrrolineN-oxide (DMPO) were conducted on an ESR5000 spectrometer (Bruker Co.). Inductively coupled plasma optical emission spectroscopy (ICP-OES) experiments were performed using an Avio 550 Max (Perkin Elmer Co.) to determine the content of metals in catalysts and leakage of metal ions during degradation. The efficiency of phenol mineralization was assessed using a total organic carbon (TOC) analyzer (Multi N/C 3100, Analytik Jena), while the phenol intermediates were identified through ultrahigh performance liquid chromatography with quadrupole time-

of-flight initial mass spectrometry (UPLC-QTOF-MS, Waters Xevo G2-XS QTOF). Furthermore, the concentration of PMS was quantified using a UV-visible spectrophotometer (AQ8100, Thermo Orion Co.) through the potassium iodide method.

## Electrochemical analysis tests

The electrochemical analysis methods linear sweep voltammetry (LSV), Cyclic Voltammetry (CV), Chronoamperometry (CA), Electrochemical impedance spectroscopy (EIS), Electrochemical active surface area (ECSA), and open circuit potential (OCPT) were shown in Supplementary Information by an electrochemical workstation (CHI 760E).

## XAFS measurements and EXAFS analysis

Chemical speciations of Cu and Ni were determined by K-edge XANES. XANES spectra of Cu and Ni in HEAs powder were obtained in the transmission model using the beamline of MEX−1 in the Australian Synchrotron Radiation Facility. Cu foil, CuO, CuPc, Ni foil, NiO, and NiPc were selected as references. The corresponding reference samples were mixed with cellulose and measured in transmission mode. Fe foil was employed for the calibration. The acquired EXAFS data were processed following standard procedures using the ATHENA (version 0.9.26) module implemented in IFEFFIT software packages for background, pre-edge line, and post-edge line calibrations. Subsequently, the $k^3$-weighted $\chi(k)$ data in the R range, spanning from 0 to 4 Å, were Fourier transformed to real (R) space using the Morlet function parameters kappaMorlet = 10 and sigmaMorlet = 2, with the assistance of hama_fortran. This transformation was carried out to distinguish the EXAFS contributions from various coordination shells.

## Theoretical calculation studies

This calculation utilizes the Cambridge Serial Total Energy Package module within the MaterialsStudio 20.1 software. The exchange-correlation functional chosen for electron-electron interactions is the Perdew-Burke-Ernzerhof functional under the Generalized Gradient Approximation. The self-consistent field (SCF) method is employed to solve the Kohn−Sham equations. The convergence criterion for the SCF energy is set to $1.0 \times 10^{-6}$ eV/atom. The total energy error of the system is within $1.0 \times 10^{-7}$ eV/atom, the stress deviation is less than 0.02 GPa, and the convergence criterion for atomic forces is 0.01 eV Å$^{-1}$. Monkhorst-Pack grids are used for Brillouin zone $k$-point sampling, with a $9 \times 9 \times 2$ $k$-point mesh chosen for calculations. The cutoff energy for the plane-wave basis set is set to 440 eV. Ultra-soft pseudopotentials are employed to describe the interaction between ionic cores and valence electrons.

## Reporting summary

Further information on research design is available in the Nature Portfolio Reporting Summary linked to this article.

# Data availability

The data supporting the findings of the study are included in the main text and supplementary information files. Additional data are available from the corresponding author upon request. Source data are provided with this paper.

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

## Acknowledgements

We acknowledge the financial support provided by the National Natural Science Foundation of China (No. 52321005), the fellowship of the Key-Area Research and Development Program of Guangdong Province (No. 2023B0101200004), the Shenzhen Science and Technology Innovation Program (No. GXWD20231129122140001, KQTD20190929172630447), the Open Project of State Key Laboratory of Urban Water Resource and Environment (Harbin Institute of Technology) (No. QA202440), the Science Foundation of National Engineering Research Center for Safe Disposal and Resources Recovery of Sludge (Harbin Institute of Technology, Grant No K2024B001). X.D. acknowledges the financial support from Australian Research Council via ARC Future Fellowship (FT230100526).

## Author contributions

Z.Y., Y.C., N.R., and X.D. designed research; Z.Y. performed research; Z.Y., K.H., Z.J., Z.S., and S.R. contributed new reagents/analytic tools; Z.Y., S.R., and X.W. analyzed data; Z.Y., Y.C., and X.D. wrote the paper.

## Competing interests

The authors declare no competing interest.
