## [Transparent Peer Review file · Nature Communications]

High-entropy alloys catalyzing polymeric transformation of water pollutants with remarkably improved electron utilization efficiency

Corresponding Author: Professor Xiaoguang Duan

Version 0:

Reviewer comments:

Reviewer #1

(Remarks to the Author)

The manuscript presents the facile and fast synthesis of high-entropy alloys and their innovative applications in AOP-mediated removal of water pollutants. The study revealed the catalyst's potential for selectively transforming micropollutants into polymeric compounds on its surface and explored the underlying sorption mechanisms through in situ Raman spectroscopy, synchrotron XAS, and DFT calculations. This selective system significantly reduces oxidant consumption while simultaneously generating valuable polymeric products. The unique configuration of the HEA fine-tunes the electronic structures of the metal centers, enhancing charge transfer capabilities and optimizing the interaction with PMS. Notably, the HEA/PMS-mediated non-radical electron transfer pathway enables ultrafast pollutant degradation over a broad pH range while exhibiting strong resistance to various real-world water interferences. The experiments are well-designed, and the mechanism interpretations are generally supported by the data. Overall, this work is suitable for publication in Nature Communications after minor revisions based on the following comments.

- 1.Except for the abbreviations used for characterizations (e.g., XANES, TEM, AC-HAADF-STEM), there are many abbreviations in the manuscript, such as PDOS, BHT, HEAs/PMS, HEAs-PMS, etc. It is essential to ensure that these abbreviations are necessary and do not confuse the readers if they only appear a couple of times.
- 2.In the abstract, some of the current keywords do not fully reflect the content's focus or key points. These keywords are crucial for accurately conveying the core message and ensuring that the abstract is easily searchable by relevant audiences. Please revise the keywords to better align with the central themes and critical aspects of the work presented in the abstract.
- 3.Line 360: The lower actual oxidant consumption compared to the theoretical oxidant consumption in AOPs suggests the occurrence of polymerization reactions. Please elaborate on this phenomenon.
- 4.Did the authors refer to any reported studies to prepare the catalysts? These references provided foundational methodologies and insights that were adapted or modified for the current research, which should be mentioned.
- 5.Line 633, 655, 680, 713, 722: The reference format is inconsistent. Please revise for uniformity and polish accordingly.
- 6.Line 252-254: "The negligible inhibitive impact of dimethyl sulfoxide (DMSO) as a specific scavenger suggests the absence of high-valence metal-oxo species in the HEAs-PMS system." Why can DMSO be used to demonstrate the role of high-valence metal-oxo species? Please explain the rationale.
- 7.The experimental conditions should be supplemented and added to the figure legends accordingly, particularly for Fig 5, S6, S10, S47.
- 8.As single atom catalysts are in recent catalysis research (also in AOPs), what are the advantages of high-entropy alloy catalysts over single-atom catalysts in catalyzing peroxide activation and generating polymeric pollutants?

Reviewer #2

(Remarks to the Author)

The manuscript reported the synthesis N-containing carbon based high-entropy alloys and its application for peroxymonosulfate activation to achieve non-mineralization phenol removal via polymeric transformation. The article has carried out a thorough exploration on the catalytic performance of HEA as well as the mechanistic study of electron transfer in PMS activation and phenol polymerization. I recommend that this paper can be accepted after the following minor

revisions:

1. The synthetic procedure of HEA depicted in Fig 1A is ambiguous. The addition/dispersion of nitrogen and carbon precursors was not involved in the diagram. Moreover, the annealing condition should be given in detail, especially the heating rate and atmosphere.
2. In traditional Fenton reactions, ROS such as $\cdot\text{OH}$, $\text{SO}_4^{\cdot-}$ and 1O_2 would generate and initiate the degradation. Why the pathway is completely suppressed and form PMS* only in the HEA-PMS system? The reason should be clarified.
3. Fe and Co were considered as the primary catalytic sites to complex with PMS for activation. But in experiments using equimolar concentrations of metal ions (Co^{2+} , Fe^{3+} , Ni^{2+} , Pd^{2+} , and Cu^{2+}) as comparative catalysts, only Co^{2+} exhibited a significant degradation performance. Why Fe^{3+} does not work? Is the degradation with Co^{2+} a polymeric transformation process or traditional ROS involved oxidation process?
4. The polymerization degree distribution analysis revealed that hexamers ($n=4$) to nonamers ($n=7$) are predominate on the HEAs surface. Would the proposition be influenced by the fact that lower-polymerization-degree products were dissolved into the solution?
5. Would the polymer generated on HEA influence its recycling? Is any additional operation required before recycling?
6. The polymerization products were generated instead of oxidized ring-opening products for further mineralization. Did the authors consider about the toxicity of the polymeric products?
7. What is the influence of initial phenol concentration in the system?

Version 1:

Reviewer comments:

Reviewer #1

(Remarks to the Author)

The authors have made appropriate revisions in accordance with the reviewers' requests, and this manuscript is well improved to satisfy the publication requirement. I agree its acceptance.

Reviewer #2

(Remarks to the Author)

The manuscript has been revised accordingly and can be accepted.

Title: *High-entropy alloys catalyzing polymeric transformation of water pollutants with remarkably improved electron utilization efficiency*

Manuscript ID: NCOMMS-24-59860

Response to Reviewers' Comments

Reviewer #1:

General comments: *The manuscript presents the facile and fast synthesis of high-entropy alloys and their innovative applications in AOP-mediated removal of water pollutants. The study revealed the catalyst's potential for selectively transforming micropollutants into polymeric compounds on its surface and explored the underlying sorption mechanisms through in situ Raman spectroscopy, synchrotron XAS, and DFT calculations. This selective system significantly reduces oxidant consumption while simultaneously generating valuable polymeric products. The unique configuration of the HEA fine-tunes the electronic structures of the metal centers, enhancing charge transfer capabilities and optimizing the interaction with PMS. Notably, the HEA/PMS-mediated non-radical electron transfer pathway enables ultrafast pollutant degradation over a broad pH range while exhibiting strong resistance to various real-world water interferences. The experiments are well-designed, and the mechanism interpretations are generally supported by the data. Overall, this work is suitable for publication in Nature Communications after minor revisions based on the following comments.*

Response: Many thanks for the Reviewer's very positive comments on the novelty and significance of this manuscript. We accept all these valuable suggestions and revised the manuscript accordingly. We hope the revision could be acceptable.

Comment 1. *Except for the abbreviations used for characterizations (e.g., XANES, TEM, AC-HAADF-STEM), there are many abbreviations in the manuscript, such as PDOS, BHT, HEAs/PMS, HEAs-PMS, etc. It is essential to ensure that these*

abbreviations are necessary and do not confuse the readers if they only appear a couple of times.

Response: Thank you for the gentle reminder. We have meticulously reviewed the manuscript and made careful adjustments to the abbreviations to improve clarity for our readers. Following the reviewer's recommendations, we have standardized abbreviations, replacing instances like "HEAs/PMS" and "NG/PMS" throughout the text with "HEAs-PMS" and "NG-PMS" to ensure clarity. Additionally, to prevent potential confusion to the readers, we have clarified the terms "PDOS" and "BHT" in the main text as follows:

Correction:

“To unravel the superior activity and electron transfer capability of HEAs, the projected density of state (PDOS) for PMS adsorption on HEAs was calculated. Initially, the PDOS of three HEAs with slight stoichiometric differences was compared.” (Page 16, Line 309-311)

“The addition of BHT(2,4,6-tri-tertbutylphenol) resulted in a notable reduction in phenol removal, demonstrating the presence of polymerization processes in both HEAs-PMS and NG-PMS systems.” (Page 18, Line 377-380)

Comment 2. *In the abstract, some of the current keywords do not fully reflect the content's focus or key points. These keywords are crucial for accurately conveying the core message and ensuring that the abstract is easily searchable by relevant audiences. Please revise the keywords to better align with the central themes and critical aspects of the work presented in the abstract.*

Response: Many thanks for your kind concern. As advised, we have changed the keywords to “Environment remediation, High-entropy alloy catalyst, Electron transfer, Oxidative polymerization pathway.”

Comment 3. *Line 360: The lower actual oxidant consumption compared to the theoretical oxidant consumption in AOPs suggests the occurrence of polymerization*

reactions. Please elaborate on this phenomenon.

Response: Many thanks for your kind enquiry. The oxidative removal of organic compounds in aqueous solution can occur through either mineralization processes (producing lower molecular weight products and ultimately CO₂) or polymerization pathways (leading to higher-molecule polymeric products) ¹⁻³. In both AOP regimes, the electrons equivalence of the oxidant (the capacity to accept electrons) should be greater than or equal to that of the reductant (total electrons that pollutants can contribute). For example, the complete mineralization of 1 mol of phenol (C₆H₅OH) to CO₂ and H₂O theoretically requires 28 mol of electron abstraction, equivalent to 14 mol of PMS. In contrast, the polymerization pathway, characterized by the generation of phenoxyl radicals via one-electron transfer only consumes 0.5 mol of PMS. This difference highlights that oxidant consumption is significantly lower in the polymerization pathway compared to mineralization processes.

Comment 4. *Did the authors refer to any reported studies to prepare the catalysts? These references provided foundational methodologies and insights that were adapted or modified for the current research, which should be mentioned.*

Response: Thanks for this great suggestion. We have carried out a comprehensive examination of the references. For HEAs preparation, we have incorporated more associated references (Nat. Commun. 11, 5437 (2020) and Sci. Bull. 67, 1890–1897 (2022)) in the revised manuscript. Building upon these studies, we optimized the synthesis method for HEAs to establish the current approach, as clarified in the main text below:

Correction:

“A straightforward one-pot oil-phase synthesis approach was adopted by heating six metal salt precursors (Cu, Pd, Fe, Co and Ni) at 220°C for 2 hours, followed by thermal decomposition along with a carbon precursor to obtain HEAs^{12,22}.” (Page 6, Line 102-105)

Comment 5. Line 633, 655, 680, 713, 722: *The reference format is inconsistent. Please revise for uniformity and polish accordingly.*

Response: Many thanks for Reviewer's kind reminder. We have revised the relevant references and conducted a thorough format check for the rest of the manuscript.

Correction:

“1. Dong, C. *et al.* Dual-functional single-atomic **Mo/Fe clusters–decorated C₃N₅** via three electron-pathway in oxygen reduction reaction for tandemly removing contaminants from water. *Proc. Natl. Acad. Sci. U.S.A.* **120**, e2305883120 (2023).

5. Chen, F. *et al.* Single-Atom Iron Anchored Tubular **g-C₃N₄ Catalysts** for Ultrafast Fenton-Like Reaction: Roles of High-Valency Iron-Oxo Species and Organic Radicals. *Adv. Mater.* **34**, 2202891 (2022).

10. He, Z. *et al.* **LaCo_{0.5}Ni_{0.5}O₃ perovskite** for efficient sulfafurazole degradation via peroxymonosulfate activation: Catalytic mechanism of interfacial structure. *Appl. Catal. B Environ.* **335**, 122883 (2023).

20. Li, B. *et al.* **Boosting N₂O Catalytic** Decomposition by the Synergistic Effect of Multiple Elements in Cobalt-Based High-Entropy Oxides. *Environ. Sci. Technol.* **58**, 2153–2161 (2024).

33. Gong, Y. *et al.* Whose Oxygen Atom Is Transferred to the Products? A Case Study of Peracetic Acid Activation via **Complexed Mn^{II} for** Organic Contaminant Degradation. *Environ. Sci. Technol.* **57**, 6723–6732 (2023).

36. Zhang, B., Li, X., Akiyama, K., Bingham, P. A. & Kubuki, S. Elucidating the Mechanistic Origin of a **Spin State-Dependent FeN_x–C Catalyst toward** Organic Contaminant Oxidation via Peroxymonosulfate Activation. *Environ. Sci. Technol.* **56**, 1321–1330 (2022).” (Page 31, References part)

Comment 6. Line 252-254: *"The negligible inhibitive impact of dimethyl sulfoxide (DMSO) as a specific scavenger suggests the absence of high-valence metal-oxo species in the HEAs-PMS system." Why can DMSO be used to demonstrate the role of*

high-valence metal-oxo species? Please explain the rationale.

Response: Many Thanks for the Reviewer's kind reminder. DMSO (dimethyl sulfoxide) is commonly used to probe high-valence metal-oxo species (e.g. Co(IV)=O, Fe(IV)=O) because of the featured oxygen transfer reaction⁴. By tracking DMSO conversion efficiency and selectivity, researchers can identify the existence and contribution of the metal-oxo species. In this HEA-PMS system, we used DMSO as a selective scavenger for metal-oxo species. Its addition did not affect the phenol removal, and thus we concluded that metal-oxo species were not generated in this system.

Comment 7. *The experimental conditions should be supplemented and added to the figure legends accordingly, particularly for Fig 5, S6, S10, S47.*

Response: Many thanks for the Reviewer's gentle reminder. We have included the experimental conditions in the captions of Fig 5, S6, S10, and S47 in the revised manuscript. Specific modifications were marked in red in the revised manuscript.

Correction:

“Fig. 5. (A) Cell viability of 16HBE under different concentrations of HEAs and Co²⁺. Cell culture time: 48 h. (B) Cells in the culture medium were observed under a microscope. (C) Treatment of phenol in real water matrices. EEM spectra of collected coal chemical industry wastewater before (D) and after treatment (E) by HEAs–PMS system. (F) Schematic of the fixed-bed column continuous flow system. (G) The phenol degradation efficiency of HEAs and annealed HEAs in cyclic experiments. (H) The efficiency of phenol removal and the metal leakage of HEAs in a continuous flow system. Dosage: [Phenol]₀: 0.1 mM, PMS: 0.25 mM, reaction solution: 50 mL, catalyst: 0.1 g/L.” (Page 25, Line 507-515)

Supplementary Information:

Fig. S6. Comparison between the apparent rate constants of HEAs and NG. Dosage: $[\text{Phenol}]_0$: 0.1 mM, PMS: 0.25 mM, reaction solution: 50 mL, catalyst: 0.1 g/L.

Fig. S11. Decomposition rate and kinetic constants of PMS with the catalytic effect of HEAs and NG. Dosage: $[\text{Phenol}]_0$: 0.1 mM, PMS: 0.25 mM, reaction solution: 50 mL, catalyst: 0.1 g/L.

Fig. S49. Cycle experiment for HEAs. Dosage: $[\text{Phenol}]_0$: 0.1 mM, PMS: 0.25 mM, reaction solution: 50 mL, catalyst: 0.1 g/L.

Comment 8. *As single atom catalysts are in recent catalysis research (also in AOPs), what are the advantages of high-entropy alloy catalysts over single-atom catalysts in catalyzing peroxide activation and generating polymeric pollutants?*

Response: Many thanks for this thought-provoking question. High-entropy alloy catalysts consist of multiple metal elements uniformly mixed in near-equal atomic ratios, creating a high level of configurational entropy. This composition provides a vast variety of active sites with different electronic microenvironments, benefited from synergistic effects among different metal elements. This contrasts with single-atom catalysts (SACs), where the catalytic activity is mainly defined by isolated and uniform metal species. The capacity to fine-tuning the electronic properties of SACs is typically limited because of the weak metal-nonmetal bonds as well as the substrate dilution effect.

In the polymerization pathway, the multi-element composition of HEAs greatly facilitates rapid electron transfer due to high conductivity and enable effective activation of peroxides because of high intrinsic activity, resulting in enhanced PMS activation in a nonradical manner to coordinate the electron-transfer pathway (ETP) with high electron utilization efficiency. Moreover, the spatial separation of active sites for PMS (HEAs) and organic pollutants (carbon substrate) in such composite structures enables continuous PMS activation, generating a higher oxidative potential for sustained ETP-mediated polymerization process. Simultaneously, the carbon support would concentrate and stabilize the polymeric intermediates, leading to the formation of high-molecular-weight polyphenolic compounds.

Reviewer #2 (Remarks to the Author):

General comments: *The manuscript reported the synthesis N-containing carbon based high-entropy alloys and its application for peroxymonosulfate activation to achieve non-mineralization phenol removal via polymeric transformation. The article has carried out a thorough exploration on the catalytic performance of HEA as well as the mechanistic study of electron transfer in PMS activation and phenol polymerization. I recommend that this paper can be accepted after the following minor revisions:*

Response: Many thanks for Reviewer's very positive comments and kind suggestions. We accepted all these valuable suggestions, which are much helpful to improve the clarification and scientific merits of the manuscript.

Comment 1. *The synthetic procedure of HEA depicted in Fig 1A is ambiguous. The addition/dispersion of nitrogen and carbon precursors was not involved in the diagram. Moreover, the annealing condition should be given in detail, especially the heating rate and atmosphere.*

Response: Many thanks to the Reviewer for the helpful reminder and kind suggestion. We have updated Figure 1A to include the addition and dispersion of nitrogen and carbon precursors to clarify the synthesis steps. Furthermore, we have provided detailed parameters, such as the heating rate and atmosphere, in the Methods section to enhance reproducibility.

Correction:

“Preparation of Catalysts. In a typical synthesis of HEAs, Pd(acac)₂ (7.6 mg), Ni(acac)₂ (6.4 mg), Fe(acac)₃ (8.8 mg), Co(acac)₃ (8.9 mg), Cu(acac)₂ (6.5 mg), **glucose (60 mg), and CTAC (50 mg) were dissolved in 10 mL of oleylamine, followed by ultrasound treatment for 30 min. Then, the mixture was heated in an oil bath at 220 °C for 2 h.** The cooled product was collected by centrifugation and washed three times with a 1:5 volume ratio ethanol/cyclohexane mixture. Finally, the black colloidal products were dispersed in cyclohexane, C₃N₄ (500 mg) was added, followed by sonication for 60 minutes, centrifugation, and washing three times with an

ethanol/cyclohexane mixture. After vacuum drying at 60°C for 12 hours, 500 mg of glucose was added and finely ground until a homogeneous dispersion was achieved. The mixture was then transferred into a tubular furnace and annealed at 800°C for 2 hours at a heating rate of 5°C/min under a protective Ar atmosphere. Subsequently, it was promptly removed and allowed to cool to room temperature to obtain HEAs. For the synthesis of NC, all the conditions are similar to those of HEAs except for the absence of black colloidal products during the annealing process.” (Page 26, Line 535-549)

Fig. 1. (A) The schematic diagram for synthesis process of HEAs

Comment 2. *In traditional Fenton reactions, ROS such as $\cdot\text{OH}$, $\text{SO}_4^{\cdot-}$ and $^1\text{O}_2$ would generate and initiate the degradation. Why the pathway is completely suppressed and form PMS* only in the HEA-PMS system? The reason should be clarified.*

Response: Thank you for your insightful question regarding the suppression of traditional reactive oxygen species (ROS) pathways in the HEA-PMS system. Based on the corresponding quenching and probe experiments (Figures 3A, S15-21), we sequentially ruled out the involvement of $\cdot\text{OH}$, $\text{SO}_4^{\cdot-}$, $^1\text{O}_2$, and high-valent metal species, in the HEA-PMS system. In contrast to conventional Fenton reactions that produce various ROS, the HEA coordinated a non-radical electron transfer pathway due to the unique electronic properties, high conductivity and spatial configuration of the high-entropy alloy active sites. The catalytic sites preferentially activate PMS to form surface-confined reactive intermediate (PMS*) with a moderate redox potential which then coordinates a single-electron transfer (SET) reaction with the co-adsorbed pollutant without generating free radicals. Additionally, the spatial separation of active sites for PMS activation (HEA) and organic pollutant adsorption (carbon substrate)

further directs the system toward an electron-transfer pathway.

Comment 3. *Fe and Co were considered as the primary catalytic sites to complex with PMS for activation. But in experiments using equimolar concentrations of metal ions (Co^{2+} , Fe^{3+} , Ni^{2+} , Pd^{2+} , and Cu^{2+}) as comparative catalysts, only Co^{2+} exhibited a significant degradation performance. Why Fe^{3+} does not work? Is the degradation with Co^{2+} a polymeric transformation process or traditional ROS involved oxidation process?*

Response: We appreciate the Reviewer's careful review and thoughtful inquiry. Typically, Fe^{3+} exhibits low activity with PMS due to its slow reaction kinetics (eq 1), and the produced $\text{SO}_5^{\cdot-}$ has a relatively low redox capacity (1.1 V)⁷⁻⁸. Although the resulting Fe^{2+} can activate PMS to produce $\text{SO}_4^{\cdot-}$ (eq 2), the process is significantly limited by the slow regeneration of Fe^{2+} , leading to an overall slow degradation efficiency of organic contaminants⁸. In the HEA-PMS system, Fe^{3+} leaching is minimal, reaching only 0.435 mg/L (Table S3), and its contributions to PMS activation and pollutant removal via homogeneous reactions are negligible (Figure S9).

As suggested, we assessed the reactive species in the Co^{2+} /PMS system (Fig. R1), which are apparently free radicals, as evidenced by the complete inhibition of phenol oxidation upon adding radical scavengers. Due to the non-selectivity and high redox potential of $\cdot\text{OH}$ (1.9-2.7 V) and $\text{SO}_4^{\cdot-}$ (2.5-3.1 V), phenol is completely mineralized (Figure R2).

Fig R1. Effect of MeOH quenchers on phenol degradation in Co^{2+} -PMS system. Conditions: $[\text{Phenol}] = 0.1 \text{ mM}$, $[\text{Co}^{2+}] = 0.49 \text{ mg/L}$, $[\text{PMS}] = 0.25 \text{ mM}$, $[\text{MeOH}] = 25 \text{ mM}$.

Fig R2. TOC removal rates in Co^{2+} -PMS system.

Comment 4. *The polymerization degree distribution analysis revealed that hexamers ($n=4$) to nonamers ($n=7$) are predominate on the HEAs surface. Would the proposition be influenced by the fact that lower-polymerization-degree products were dissolved into the solution?*

Response: We sincerely appreciate the Reviewer's insightful question. We agree that the distribution of products is influenced by the degree of polymerization (molecular size). As shown in Fig. S36, significant amounts of low-polymerization-degree products ($n = 0-2$), ranging from dimers to tetramers, were detected in the solution phase of the HEAs-PMS system. This is attributed to the rapid phenol oxidation kinetics in the HEAs-PMS system, where some low-polymerization products are released into the solution phase before undergoing further polymerization. Moreover, the surface affinity of organic intermediates via π - π interactions generally decreases as the molecular size decreases, resulting in a higher concentration of lower-polymerization-degree products dissolving into the solution. Therefore, additional details regarding this aspect have been incorporated into the revised manuscript.

Fig. S36. UPLC–QTOF-MS chromatograms of phenol oxidation polymerization products in the HEAs/PMS system solution (min), and their corresponding molecular ion mass spectra of the chromatographic peaks.

Correction:

“Moreover, upon the depletion of the electron source (PMS), the insufficient coupling of polymer radicals can lead to the termination of polymerization reactions. This result is consistent with the findings of the TGA experiment that HEAs exhibit higher pollutant-to-polymer transformation efficiency. In **Fig. 4E**, polymerization degree distribution analysis on the surfaces of NG and HEAs revealed that tetramers ($n=2$) and pentamers ($n=3$) are predominantly observed on the NG surface, **while most hexamers ($n=4$) to nonamers ($n=7$) are present on the HEAs surface. In contrast, lower-polymerization-degree products are dissolved in the solution phase.** Thus, the higher redox potential of HEAs-PMS* and faster electron transfer capability can trigger the rapid generation and accumulation of phenoxyl radicals, thereby enhancing the phenol-to-polymer transformation kinetics to yield larger polyphenol products.” (Page 20, Line 415-425)

Comment 5. *Would the polymer generated on HEA influence its recycling? Is any*

additional operation required before recycling?

Response: We thank the Reviewer for the insightful questions. To determine whether the polymeric products influence the recycling of the catalyst, we conducted cyclic tests on the HEA catalyst to evaluate its stability and performance during repeated phenol degradation cycles. The results showed that after five consecutive runs, the phenol removal rate remained above 95%, demonstrating the catalyst's superior stability during cyclic degradation (Figure S47). Although the accumulation of polymeric products on the catalyst surface led to a gradual slight decrease in the reaction rate, the polymers did not significantly impact the exposure of active sites, allowing continuous PMS activation. This is because the catalyst primarily activates PMS on HEA nanoparticles, while the polymer products mainly form on the graphite surface. Thus, the spatial separation of PMS activation and polymerization processes ensured the high stability of the catalyst, allowing for sustained PMS activation and a thermodynamically favorable environment for continuous polymerization. To regenerate the catalyst, tetrahydrofuran (THF) and toluene can be used as solvents to dissolve and remove the organic polymers accumulated on the catalyst surface, partially restoring catalyst activity, as shown in Figure R3.

Fig R3. The phenol removal performance of degraded HEAs and HEAs after the surface polymers were eluted with toluene. Conditions: [Phenol] = 0.1 mM, [catalyst] = 0.1 g/L, [PMS] = 0.25 mM.

Comment 6. *The polymerization products were generated instead of oxidized ring-opening products for further mineralization. Did the authors consider about the toxicity of the polymeric products?*

Response: We appreciate the Reviewer's consideration regarding the toxicity of the polymeric products. As suggested, we conducted a detailed analysis of the polymer products formed in the HEAs-PMS system using the Toxicity Estimation Software Tool (TEST). Specifically, we used the Developmental Toxicity module in the TEST software to evaluate the toxicity of dimers to tetramers. The results indicated that the developmental toxicity remained relatively stable, with only a slight increasing trend as the degree of polymerization increased. These findings highlight the importance of effectively recovering the solid polymeric products from the catalyst surface to minimize potential environmental impacts and utilize the upcycled products in a sustainable manner.

Fig. S47. Developmental toxicity at varying degrees of polymerization calculated using the Toxicity Estimation Software Tool (TEST).

Correction:

“Notably, the addition of HEAs yielded promising results, maintaining 16HBE cell line viability above 90% even with twice the concentration. This contrast between HEAs and ions was further emphasized in **Fig. 5B**, where a noticeable difference in cell count and survival morphology in the microscopic field. **In addition, the toxicity of the polymeric products was assessed using the Toxicity Estimation Software Tool (TEST), which showed a slight increase in toxicity as the polymerization degree increased. This**

finding highlights the importance of developing efficient recovery technologies to reduce potential environmental impacts and promote the reutilization the upcycled polymeric products.” (Page 23, Line 478-486)

Comment 7. *What is the influence of initial phenol concentration in the system?*

Response: We appreciate the Reviewer's insightful question regarding the influence of the initial phenol concentration. We conducted experiments with various initial phenol concentrations, as presented in Fig. S8. As expected, at a fixed PMS loading of 0.25 mM, the phenol removal rate significantly decreased as the initial phenol concentration increased. However, complete phenol removal was still achieved within 10 minutes at an initial concentration of 0.1 mM. When the initial phenol concentration was higher (0.5 mM) and exceeded the electron equivalents that PMS could provide (0.25 mM), the reaction rate decreased to 0.008 min^{-1} .

It is worth noting that a higher pollutant concentration generally benefits the polymerization process, as it continuously supplies monomer precursors for chain-growing, cross-linking, and coupling reactions. As shown in Figure R4, the absolute amount of phenol removed (in mM) increased with higher initial concentrations. Within the effective electron utilization range of PMS, increasing the pollutant concentration helps elevate the surface organic radical concentration, thereby enhancing the polymerization reaction. However, excessively high phenol concentrations can result in excess phenol adsorbing onto the catalyst surface, hindering PMS adsorption. Therefore, optimizing the PMS and catalyst dosages is crucial for achieving complete pollutant removal with high efficiency, particularly at elevated initial pollutant concentrations.

Fig S8. The phenol removal performance of different initial phenol concentration. Conditions: [catalyst] = 0.1 g/L, reaction solution: 50 mL, [PMS] = 0.25 mM, reaction time: 60 min.

Fig R4. The conversion of phenol at varying initial concentrations. Conditions: [catalyst] = 0.1 g/L, reaction solution: 50 mL, [PMS] = 0.25 mM, reaction time: 60 min.

Correction:

“Additionally, the optimization of PMS concentration demonstrated that complete degradation was achievable with a PMS dose of 0.25 mM, following a positive correlation of oxidation kinetics with PMS concentration, likely due to its positive feedback effects on PMS activation (*Supplementary Fig. S7*). The removal rates decreased with increasing initial phenol concentrations, though complete phenol removal was still achieved within 10 min at a concentration of 0.1 mM (*Supplementary*

Fig. S8). However, when the initial phenol concentration exceeded the electron equivalents that PMS could accept (0.25 mM), the removal rate was significantly reduced.” (Page 9, Line 160-168)

References

1. Zhang, Y.-J. *et al.* Simultaneous nanocatalytic surface activation of pollutants and oxidants for highly efficient water decontamination. *Nat. Commun.* **13**, 3005 (2022).
2. Zhang, Y.-J. *et al.* Distinguishing homogeneous advanced oxidation processes in bulk water from heterogeneous surface reactions in organic oxidation. *Proc. Natl. Acad. Sci. U.S.A.* **120**, e2302407120 (2023).
3. Liu, H.-Z. *et al.* Tailoring d-band center of high-valent metal-oxo species for pollutant removal via complete polymerization. *Nat. Commun.* **15**, 2327 (2024).
4. Deng, G. *et al.* Ferryl Ion in the Photo-Fenton Process at Acidic pH: Occurrence, Fate, and Implications. *Environ. Sci. Technol.* **57**, 18586–18596 (2023).

Reviewers' Comments

Reviewer #1(Remarks to the Author):

The authors have made appropriate revisions in accordance with the reviewers' requests, and this manuscript is well improved to satisfy the publication requirement. I agree its acceptance.

Reviewer #2 (Remarks to the Author):

The manuscript has been revised accordingly and can be accepted.